# Terconazole, an Azole Antifungal Drug, Increases Cytotoxicity in Antimitotic Drug-Treated Resistant Cancer Cells with Substrate-Specific P-gp Inhibitory Activity

**DOI:** 10.3390/ijms232213809

**Published:** 2022-11-09

**Authors:** Ji Sun Lee, Yunmoon Oh, Jae Hyeon Park, So Young Kyung, Hyung Sik Kim, Sungpil Yoon

**Affiliations:** School of Pharmacy, Sungkyunkwan University, 2066 Seobu-ro, Jangan-gu, Suwon 16419, Korea

**Keywords:** azole antifungal drugs, terconazole, co-treatment, cancer, P-gp, drug resistance

## Abstract

Azole antifungal drugs have been shown to enhance the cytotoxicity of antimitotic drugs in P-glycoprotein (P-gp)-overexpressing-resistant cancer cells. Herein, we examined two azole antifungal drugs, terconazole (TCZ) and butoconazole (BTZ), previously unexplored in resistant cancers. We found that both TCZ and BTZ increased cytotoxicity in vincristine (VIC)-treated P-gp-overexpressing drug-resistant KBV20C cancer cells. Following detailed analysis, low-dose VIC + TCZ exerted higher cytotoxicity than co-treatment with VIC + BTZ. Furthermore, we found that VIC + TCZ could increase apoptosis and induce G2 arrest. Additionally, low-dose TCZ could be combined with various antimitotic drugs to increase their cytotoxicity in P-gp-overexpressing antimitotic drug-resistant cancer cells. Moreover, TCZ exhibited P-gp inhibitory activity, suggesting that the inhibitory activity of P-gp plays a role in sensitization afforded by VIC + TCZ co-treatment. We also evaluated the cytotoxicity of 12 azole antifungal drugs at low doses in drug-resistant cancer cells. VIC + TCZ, VIC + itraconazole, and VIC + posaconazole exhibited the strongest cytotoxicity in P-gp-overexpressing KBV20C and MCF-7/ADR-resistant cancer cells. These drugs exerted robust P-gp inhibitory activity, accompanied by calcein-AM substrate efflux. Given that azole antifungal drugs have long been used in clinics, our results, which reposition azole antifungal drugs for treating P-gp-overexpressing-resistant cancer, could be employed to treat patients with drug-resistant cancer rapidly.

## 1. Introduction

Antimitotic drugs, such as vincristine (VIC), eribulin, vinorelbine, vinblastine, paclitaxel, and docetaxel, can inhibit cellular division by preventing chromosomal segregations that occur via microtubule polymerization or depolymerization [1,2,3,4]. Multidrug resistance (MDR) has been reported in patients administered antimitotic agents [4,5,6,7]. Investigations to identify novel drugs and mechanisms underlying MDR are crucial to facilitate rapid and more effective treatment strategies in patients with MDR cancer.

P-glycoprotein (P-gp) is located in the cellular membrane and is responsible for the efflux of anticancer drugs [8]. P-gp overexpression is a well-known mechanism underlying MDR in cancer cells [8,9]. Attempts are ongoing to develop P-gp inhibitors with reduced toxicity toward normal cells, capable of specifically targeting drug-resistant cancer cells [6,10,11]. Identifying mechanisms capable of sensitizing P-gp-overexpressing cancer cells could improve treatment options in patients who develop resistance to antimitotic drugs. Although P-gp inhibitors have been developed, their toxicity toward normal cells results in treatment failure [12,13,14]. Therefore, it is important to establish novel therapeutic options that specifically target P-gp-overexpressing, drug-resistant cancer cells.

We have previously identified novel mechanisms and inhibitors that target P-gp activity with low toxicity toward normal cells [15,16,17,18,19]. Therefore, we were interested in the repositioning of the US Food and Drug Administration (FDA)-approved drugs with known toxicity [20,21,22]. On identifying instances of novel FDA-approved drug repositioning for sensitization of P-gp-overexpressing-resistant cancers, we speculate that these drugs could be administered to patients with MDR cancer without warranting additional toxicity testing. Thus, the urgent need for pharmacological treatments against P-gp-overexpressing-resistant cancer can be effectively addressed if novel mechanisms of approved anticancer drugs are identified, given that these drugs can be used without further toxicity assessments [21,22,23].

Azole antifungal drugs (e.g., itraconazole, econazole, and ketoconazole) reportedly exhibit anticancer activity [24,25,26,27,28]. In addition, these agents could exert P-gp inhibitory activity, and their combination with chemotherapeutic drugs could increase cytotoxicity in resistant cancers. Moreover, these agents have been repositioned in clinical trials to treat patients with cancer [26,27,29,30]. However, terconazole (TCZ) and butoconazole (BTZ) have not been examined for targeting drug-resistant cancer cells.

In the present study, we examined whether TCZ, an FDA-approved drug, increases the cytotoxicity of antimitotic drugs in P-gp-overexpressing MDR cancer cells. We also determined the cytotoxic mechanisms mediated by these drugs in combination with antimitotic agents. Finally, we compared the efficacy of 12 azole antifungal drugs at low doses to identify their potential superior efficacy in sensitizing drug-resistant cancer cells. In particular, we investigated whether azole antifungal drugs could inhibit P-gp activity in MDR cancer cells using two different P-gp substrates, rhodamine 123 and calcein-AM. Given the introduction of personalized medicine, our results may facilitate the development of azole antifungal drug-based therapies against P-gp-overexpressing drug-resistant cancers.

## 2. Results

### 2.1. Both TCZ and BTZ Increase VIC-Induced Cytotoxicity toward P-gp-Overexpressing Drug-Resistant KBV20C Cancer Cells

We examined whether low-dose azole antifungal drugs could increase the cytotoxicity of antimitotic drugs in drug-resistant KBV20C cancer cells. The potential of TCZ and BTZ has not been explored in resistant cancers, and their relative cytotoxic effects in comparison with other azole antifungal drugs have not been established. Therefore, we determined the cytotoxicity of low-dose TCZ and BTZ in VIC-treated P-gp-overexpressing drug-resistant KBV20C cancer cells.

As shown in Figure 1A–C, co-treatment with 5 μM TCZ or BTZ markedly reduced proliferation in VIC-treated KBV20C cells. However, single-agent treatment with 5 μM TCZ or BTZ failed to exert cytotoxicity against P-gp overexpression in both resistant KBV20C and sensitive KB cells (Figure 1A,D). Following a more detailed analysis, we found that low-dose TCZ exhibited greater cytotoxicity than BTZ (Figure 1A,B). Recently, we have shown that the ansamycin antibiotic rifabutin could increase cytotoxicity in P-gp-overexpressing-resistant cancers [18]. Both TCZ and BTZ exhibited fewer cytotoxic effects than an equivalent dose of rifabutin (positive control). (Figure 1A,E) We also compared the cytotoxicity of TCZ with that of verapamil, a well-known P-gp inhibitor. As shown in Figure 1E, TCZ (5 μM) and verapamil (10 μM) induced similar sensitization effects in VIC-treated cells, suggesting that low-dose TCZ could increase cytotoxicity in VIC-treated KBV20C cells similar to verapamil for sensitizing P-gp-overexpressing-resistant KBV20C cells.

### 2.2. Co-Treatment with VIC and TCZ Reduces Long-Term Survival as Determined by Colony-Forming Assays

Next, we evaluated whether low-dose TCZ could sensitize drug-resistant KBV20C cancer cells to induce long-term survival. We performed colony-forming assays 10 d after drug treatment.

As seen in Figure 2A, 2.5 μM and 5 μM TCZ reduced colony formation in VIC-treated KBV20C cells, whereas single-agent treatment with VIC or TCZ showed similar size and numbers to those at control levels. These results suggested that combined therapy with VIC and TCZ could be employed to treat VIC-resistant cancer cells for long-term efficacy. We found that 2.5 μM TCZ sufficiently increased cytotoxicity in P-gp-overexpressing KBV20C cells (Figure 2A). This finding indicated that low-dose TCZ could adequately increase cytotoxicity when combined with VIC.

### 2.3. TCZ Dose- and Time-Dependently Increases Late Apoptosis in VIC Co-Treated KBV20C Cells

To comprehensively clarify the cytotoxic mechanism of VIC + TCZ co-treatment, apoptotic cell death was analyzed using annexin V staining. We quantitatively estimated the number of apoptotic cells distributed in the early and late stages. As shown in Figure 2B, early apoptotic cells proportionally increased with the increasing TCZ concentration. Treatment with 2.5 and 5 μM TCZ increased early apoptotic cell death by 11 and 15%, respectively (Figure 2B). Based on these findings, early apoptosis in VIC + TCZ-co-treated cells could be increased by increasing the TCZ concentration. However, we also noted a slight increase in late apoptotic death with increasing TCZ doses (Figure 2B), suggesting that induction of early apoptosis could induce the cytotoxic effects of VIC + TCZ. We assumed that VIC + TCZ could increase late apoptosis by increasing the time interval. Subsequently, we examined 48 h treatment with VIC + TCZ co-treatment. We found that treatment for 48 h dose-dependently increased late apoptosis (Figure 2C), suggesting that increased early apoptotic cells stimulate cellular death. Considering that early apoptosis could induce late apoptosis within a relatively short time interval, we concluded that co-treatment with VIC + TCZ increased cytotoxicity via non-delayed apoptosis for the recovery of resistant cancer cells. We measured the expression level of cleaved poly-ADP-ribose polymerases (C-PARP), a well-known apoptotic marker, to confirm the increased apoptosis in VIC + TCZ-co-treated cells at the molecular level [15,16,31]. As shown in Figure 2D, VIC + TCZ-co-treated cells showed increased C-PARP production. In the densitometric analysis, co-treatment resulted in two-fold higher C-PARP levels than treatment with a single agent. Taken together, our results demonstrated that TCZ could significantly increase the cytotoxicity of VIC toward P-gp-overexpressing drug-resistant KBV20C cancer cells by inducing apoptosis.

### 2.4. TCZ Induces G2-Arrest in VIC co-Treated Resistant KBV20C Cells

Next, we performed fluorescence-activated cell sorting (FACS) analysis to determine whether cell cycle arrest was involved in the early apoptotic cell death after VIC + TCZ co-treatment. As shown in Figure 3A, G2 arrested cells proportionally increased with increasing TCZ doses. These results suggested that VIC + TCZ-induced G2 arrest could be enhanced with increasing TCZ concentrations. We performed western blot analysis to further examine the expression of proteins involved in G2 arrest [15,16,31]. As seen in Figure 2D, VIC + TCZ co-treatment increased the expression of cyclin B1 protein. In the densitometric analysis, pAkt levels were similar between co-treatment and single-agent treatments.

### 2.5. TCZ Exhibits Low P-gp-Inhibitory Activity in the Rhodamine 123 Efflux Assay

To elucidate the potential mechanisms mediating the effects of VIC + TCZ, we first analyzed the P-gp inhibitory activity of TCZ in P-gp-overexpressing KBV20C cancer cells. We postulated that TCZ-induced P-gp inhibition mediated a cytotoxic effect in VIC-treated KBV20C cells. Previously, aripiprazole and reserpine have shown potent P-gp inhibitory activity [16,18,19,31]; hence, these agents were used as positive controls for comparing the effects of TCZ. We determined the number of KBV20C cells accumulating rhodamine 123, a well-known P-gp substrate, to measure P-gp inhibition [16,18,19,31].

As shown in Figure 3B, the positive controls (aripiprazole and reserpine) increased P-gp inhibitory activity by approximately 500–600% in both 4- and 24-h drug treatments when compared with dimethyl sulfoxide (DMSO)-treated controls. However, low-dose TCZ afforded markedly low P-gp inhibitory activity, with less than a 200% increase, thereby indicating that P-gp inhibition may fail to play a major role in sensitizing P-gp-overexpressing-resistant cancer cells to VIC. However, despite the low P-gp-inhibitory activity, TCZ could sensitize VIC-treated KBV20C cells; hence, this co-treatment strategy could be useful in clinical settings owing to the minimal toxic P-gp-inhibitory effects on normal cells. Collectively, low-dose treatment with TCZ could increase the cytotoxicity of VIC in P-gp-overexpressing KBV20C cells via weak P-gp inhibition of P-gp efflux.

### 2.6. Co-Treatment with TCZ Increases Cytotoxicity of Other Antimitotic Drugs-Treated KBV20C Cells

Next, we determined whether co-treatment with TCZ could increase the cytotoxicity of other antimitotic drugs. We examined the cytotoxic effects of TCZ in combination with eribulin, vinorelbine, and docetaxel, antimitotic drugs used as chemotherapeutic agents in cancer patients [1,2,3]. As shown in Figure 3C, eribulin + TCZ, vinorelbine + TCZ, and docetaxel + TCZ showed increased cytotoxic effects against P-gp-overexpressing-resistant KBV20C cells. As shown in Figure 3D, eribulin + TCZ, vinorelbine + TCZ, and docetaxel + TCZ markedly increased early apoptosis. These results confirmed that eribulin + TCZ, vinorelbine + TCZ, and docetaxel + TCZ exerted cytotoxicity quantitatively similar to that of VIC + TCZ co-treatment in sensitizing drug-resistant KBV20C cancer cells. Overall, this result suggested that TCZ could be combined with other antimitotic drugs to sensitize P-gp-overexpressing-resistant cancer cells.

### 2.7. TCZ, Itraconazole, and Posaconazole at Low Doses Exhibit Higher Cytotoxic Effects in P-gp-Overexpressing Cancer Cells Than Those of Other Azole Antifungal Drugs

We further examined whether P-gp-overexpressing drug-resistant KBV20C cells could be sensitized by co-treatment with other azole antifungal drugs [32,33]. Although previous studies have shown that co-treatments induce cytotoxic effects in P-gp-overexpressing drug-resistant cancer cells [28,34], low-dose azole treatments have not been well-evaluated. As shown in Figure 4A–C, three azoles (itraconazole, posaconazole, and TCZ) at low doses exhibited the strongest cytotoxic effects in P-gp-overexpressing-resistant KBV20C cells. Co-treatment with econazole, ketoconazole, oxiconazole, BTZ, sulconazole, and sertaconazole also showed sensitization effects in VIC-treated KBV20C cells (Figure 4A–C). However, miconazole, voriconazole, and fluconazole demonstrated substantially lower cytotoxic effects (Figure 4D,E). Collectively, among the 12 azole antifungal drugs examined, low-dose co-treatment with TCZ, itraconazole, and posaconazole was highly cytotoxic in VIC-treated resistant KBV20C cancer cells.

### 2.8. Co-Treatment with TCZ Increases Cytotoxicity in Other P-gp-Overexpressing-Resistant MCF-7/ADR Cancer Cells

We further determined whether TCZ could sensitize other P-gp-overexpressing-resistant cancer cells to confirm that VIC + TCZ can be generally employed for P-gp-overexpressing drug-resistant cancer cell types. We evaluated MCF-7/ADR-resistant breast cancer cells, routinely studied for analyzing P-gp-overexpressing-resistant cancer [35,36]. Previously, MCF-7/ADR was found to exhibit considerably increased P-gp expression levels when compared with P-gp-overexpressing-resistant KBV20C cells [37]. As shown in Figure 5A, TCZ, itraconazole, and posaconazole increased cytotoxicity in VIC-treated MCF-7/ADR-resistant cancer cells. As a positive control, we compared the ansamycin antibiotic rifabutin with the same dose of azole antifungal drugs [18]. We observed that all three azole antifungal drugs (TCZ, itraconazole, and posaconazole) and rifabutin exerted similar levels of cytotoxicity in VIC-treated MCF-7/ADR cells. This finding suggested that TCZ could generally increase cytotoxicity in P-gp-overexpressing cancer cells.

### 2.9. TCZ, Itraconazole, and Posaconazole Exert High P-gp Inhibitory Activity in Calcein-AM Substrate Efflux Assays

Next, we compared the P-gp inhibitory activity of TCZ, itraconazole, and posaconazole in P-gp-overexpressing KBV20C and MCF-7/ADR cells. We hypothesized that P-gp inhibition by these azole antifungal drugs might be responsible for their sensitizing effects in VIC-treated KBV20C and MCF-7/ADR drug-resistant cells.

Verapamil, a P-gp inhibitor, was used as the positive control and has been shown to increase the inhibitory activity of the P-gp substrate and rhodamine 123 efflux [16,17,18,19]. As shown in the left panels in Figure 5B,C, we observed approximately 300% higher P-gp inhibitory activity in verapamil-treated KBV20C cells than in the DMSO-treated control KBV20C cells; however, verapamil largely inhibited P-gp activity in MCF-7/ADR cells, with a 1000% increase in inhibitory activity. These findings indicated that KBV20C and MCF-7/ADR cells exhibit distinct P-gp overexpression levels and distributions in cellular membranes. We observed a 200% and 400% increase in TCZ-mediated P-gp inhibition in KBV20C and MCF-7 ADR cells, respectively (left panels in Figure 5B,C). However, we did not detect inhibitory activity in itraconazole- and posaconazole-treated cells, similar to DMSO-treated control levels (left panels in Figure 5B,C). These findings indicated that TCZ exerted a stronger P-gp inhibitory activity with efflux of rhodamine 123 substrates than that of itraconazole and posaconazole, despite inducing similar sensitization effects in P-gp-overexpressing KBV20C and MCF-7/ADR cells.

Furthermore, we investigated P-gp inhibitory activity using calcein-AM, another well-known P-gp substrate, to measure P-gp inhibition [38,39]. As observed in rhodamine 123 experiments, yellow fluorescence in the cell indicated intracellular accumulation of calcein-AM lasting for 3 h. Verapamil showed high inhibitory activity on calcein-AM efflux in MCF-7/ADR cells, with more than a 2000% increase when compared with DMSO-treated control levels (right panel in Figure 5C). Surprisingly, using calcein-AM, we found that itraconazole and posaconazole exhibited higher P-gp inhibitory activity, whereas TCZ demonstrated lower P-gp inhibitory activity than the positive control verapamil in both KBV20C and MCF-7/ADR cells (right panels in Figure 5B,C). The results indicate that TCZ, itraconazole, and posaconazole exhibited strong P-gp-inhibitory activity with calcein-AM-substrate specificity in both KBV20C and MCF-7/ADR cells.

In summary, TCZ increased cytotoxicity in P-gp-overexpressing-resistant cancer cells via P-gp inhibition. Furthermore, TCZ, itraconazole, and posaconazole exerted distinct substrate-specific inhibition of P-gp, despite being azole antifungal drugs. Considering the high inhibitory activity on calcein-AM substrates, we concluded that TCZ, itraconazole, and posaconazole exhibit high substrate specificity for preventing P-gp efflux. In addition, given the popularity of personalized medicine in resistant cancer therapy, our results could contribute to therapeutic prescriptions for P-gp-overexpressing-resistant cancer types. For example, TCZ, itraconazole, and posaconazole could target P-gp-overexpressing drug-resistant cancers specific for rhodamine 123 or calcein-AM efflux. Accordingly, low-dose treatment with TCZ, itraconazole, and posaconazole may be useful in clinical settings for various P-gp-overexpressing-resistant cancer types.

## 3. Discussion

Azole antifungal drugs have been suggested for resistant cancer therapy and have shown inhibitory activity against P-gp [25,28,29,30,34]. For example, itraconazole, ketoconazole, and econazole reportedly enhance cytotoxicity in P-gp-overexpressing drug-resistant cancer cells [25,26,27,30]. Given that azole drugs have long been used in clinical settings, they could be easily employed in patients with cancer if their cancer-targeting cytotoxicity can be established in resistant cancers.

Therefore, we searched for novel azole antifungal drugs capable of increasing the cytotoxicity of antimitotic drugs against P-gp-overexpressing drug-resistant cancers. In a detailed quantitative analysis assessing novel azole antifungal drugs, we found that low-dose TCZ was more effective than BTZ for sensitizing VIC-treated resistant KBV20C cells. Microscopy, FACS, and annexin V analyses revealed that treatment with TCZ could upregulate apoptosis, resulting from increased G2 arrest and decreased proliferation of P-gp-overexpressing drug-resistant KBV20C cells. Future investigation should explore whether VIC + TCZ can increase autophagy or DNA damage in in-depth mechanism studies. Further in vivo studies using animal models are needed to facilitate the rapid application of TCZ in patients with MDR. Considering that TCZ could sensitize VIC-treated KBV20C cells at low doses, it may be useful in clinical settings, given its minimal toxicity in normal cells. We believe that our findings will facilitate the application of TCZ in P-gp-overexpressing-resistant patients, affording a potential combination therapy with antimitotic drugs. Moreover, we demonstrated that TCZ exhibited sensitization effects similar to those of eribulin-, vinorelbine-, and docetaxel-treated resistant KBV20C cells. We hypothesized that TCZ can be co-administered with various anticancer drugs to sensitize MDR cancer cells. Co-treatment with VIC + TCZ induced marked cytotoxicity in P-gp-overexpressing cancer cells, along with similar sensitization effects in highly P-gp-overexpressing MCF-7/ADR breast cancer cell types.

Previously, azole antifungal drugs have been used to sensitize P-gp-overexpressing-resistant cancer cells [25,26,27,28,29,30,34]. However, comparisons of individual azole antifungal drugs and their exact mechanisms of action have not been comprehensively clarified in drug-resistant, P-gp-overexpressing cancer cells. We performed a search of the literature and identified 12 azole antifungals [32,33]. Some of these drugs have previously demonstrated anticancer effects, along with detailed mechanisms of action [24,25,29,30]. Therefore, clinical trials have evaluated their efficacy in treating solid tumors [27,28,34]. Among the 12 azole antifungal drugs examined, co-treatment with three drugs, i.e., TCZ, itraconazole, and posaconazole, could highly sensitize antimitotic drug-resistant KBV20C and MCF-7/ADR cells at relatively low doses. Although the resistant cancer-sensitizing abilities of azole antifungal drugs have been previously demonstrated [26,27,28,30,34], our findings represent a pioneering and effective application of 12 azole antifungal drugs as potential repurposing candidates.

The efflux of VIC by P-gp is the primary mechanism underlying the KBV20C cell resistance to VIC [16,19,31]. We examined whether the cytotoxicity of VIC + TCZ, VIC + itraconazole, and VIC + posaconazole co-treatments could be attributed to their P-gp-inhibitory effects. When rhodamine 123 was used as the substrate efflux assay, verapamil (positive control) treatment afforded 3- and 10-fold higher inhibitory activities in KBV20C and MCF-7/ADR cells, respectively, than those with DMSO treatment. MCF-7/ADR cells exhibit markedly higher P-gp-overexpression in the cellular membranes than that in the KBV20C cells [37]. However, we detected low or non-P-gp inhibition with TCZ, itraconazole, and posaconazole. The inhibitory activity of TCZ was approximately two- and four-fold in KBV20C and MCF-7/ADR cells, whereas inhibitory activities of itraconazole and posaconazole were similar to DMSO treatment in both resistant cancer cell lines. This suggests that TCZ, itraconazole, and posaconazole can sensitize P-gp-overexpressing-resistant cancer cells through alternate mechanisms, such as removing or inhibiting factors that block the effect of VIC in drug-resistant cancer cells, with co-treatment affording a synergistic effect on cells.

Given that itraconazole has previously shown minimal P-gp-inhibitory activity using rhodamine 123 [28], we tested efflux with another substrate, calcein-AM [38,39], in P-gp-overexpressing cancer cells. Verapamil, a positive control, demonstrated high inhibitory activity against calcein-AM efflux in MCF-7/ADR cells, approximately 20-fold greater than that of DMSO in control cells. Surprisingly, with calcein-AM as the substrate, itraconazole and posaconazole showed high P-gp-inhibitory activities in both KBV20C and MCF-7/ADR cells. This finding indicates that the P-gp inhibitory activity of TCZ, itraconazole, and posaconazole primarily contributes to their highly cytotoxic effect on P-gp-overexpressing drug-resistant KBV20C and MCF-7/ADR cancer cells treated with antimitotic drugs.

Additionally, we compared the inhibitory activity of TCZ, itraconazole, and posaconazole on calcein-AM and found that TCZ exerted lower inhibition than that of itraconazole and posaconazole. Considering that TCZ exhibits greater P-gp inhibitory activity against rhodamine 123 than that of itraconazole and posaconazole, the P-gp inhibitory mechanisms of TCZ could be distinct from those of other azole antifungal drugs. Personalized medicine is gaining momentum, and our results indicate that azole antifungal drugs may be valuable in improving the effectiveness of current prescriptions in patients with drug-resistant cancer. These patients are generally allergic or sensitive to P-gp-inhibitory effects in normal tissues. Further investigation of these drugs is warranted to determine the molecular targets for sensitizing resistant cancer cells without P-gp inhibition. Further assessment of sensitization using ultralow attachment spheroids or droplet assay will facilitate the rapid application of TCZ, especially in patients resistant to combination therapy with antimitotic drugs.

Given that TCZ is an FDA-approved drug and its toxicity is well documented in clinics, it can be efficiently used to address the urgent need for the pharmacological treatment of antimitotic drug-resistant cancers. In addition to itraconazole, TCZ may be beneficial in treating P-gp-overexpressing-resistant patients in clinical trials.

## 4. Materials and Methods

### 4.1. Reagents and Cell Culture

Rhodamine 123 and verapamil were purchased from Sigma-Aldrich (St. Louis, MO, USA). Calcein-AM was purchased from Invitrogen (St. Louis, MO, USA). VIC and vinorelbine were purchased from Enzo Life Sciences (Farmingdale, NY, USA). TCZ, BTZ nitrate, itraconazole, posaconazole, econazole nitrate, ketoconazole, oxiconazole nitrate, sulconazole, sertaconazole, miconazole, voriconazole, fluconazole, aripiprazole, reserpine, and rifabutin were purchased from Selleckchem (Houston, TX, USA). Both aqueous solutions of eribulin (Eisai Korea, Seoul, South Korea) and docetaxel (Aventis, Bridgewater, NJ, USA) were obtained from the National Cancer Center in South Korea. Antibodies against C-PARP, pAkt, and Cyclin B1 were obtained from Cell Signaling Technology (Danvers, MA, USA). Antibodies against Cyclin D1, p21, and GAPDH were obtained from Santa Cruz Biotechnology (Santa Cruz, CA, USA).

The human oral squamous carcinoma cell line, KB, and its multidrug-resistant sub-line, KBV20C, have been previously described [18,19,40]. P-gp-overexpressing-resistant MCF-7/ADR breast cancer cells were generally gifted from Dr. Yong Kee Kim (College of Pharmacy, Sookmyung Women’s University, Seoul, South Korea) and have been previously used [35,36]. All cell lines were cultured in RPMI 1640 or DMEM containing 10% fetal bovine serum, 100 U/mL penicillin, and 100 μg/mL streptomycin (WelGENE, Daegu, South Korea).

### 4.2. Microscopic Observation

Cellular growth was observed with a microscope as previously described [19,41,42]. Briefly, KBV20C, MCF-7/ADR, or KB cells were grown in 60-mm-diameter dishes and treated with 5 nM VIC, 60 nM eribulin, 0.5 μg/mL vinorelbine, 1 μg/mL docetaxel, 2.5 (or 5 μM) azole antifungal drugs, 2.5 (or 5 μM) rifabutin, alone or in combination with antimitotic drugs (VIC, eribulin, vinorelbine, or docetaxel), or 0.1% DMSO (control) for 1 day. These cells were then examined immediately in two independent experiments using an ECLIPSETs2 inverted routine microscope (Nikon, Tokyo, Japan) with a ×40 or a ×100 objective lens. They were qualitatively observed and confirmed by performing at least two independent experiments.

### 4.3. Cell Viability Assay

Cell proliferation was measured by a colorimetric assay using an EZ-CyTox cell viability assay kit (Daeillab, South Korea) as previously described [19,41,42]. Briefly, KBV20C cells were plated on 96-well plates and grown to 20–30% confluence. The cells were then treated for 48 h with 5 nM VIC, 5 μM TCZ, 5 μM BTZ, 5 μM rifabutin, alone or in combination with 5 nM VIC, or 0.1% DMSO (control). They were then incubated with 10 μL EZ-CyTox solution for 1 h at 37 °C. The absorbance at 450 nm was measured using the VERSA MAX Microplate Reader (Molecular Devices Corp., Sunnyvale, CA, USA). All experiments were performed at least in triplicate and repeated twice.

### 4.4. Colony-Forming Assay

A colony-forming assay was used to assess long-term growth in the presence of a drug based on a previously described method [18,19,31]. Briefly, 1–2 × 10^3^ cells were grown in 6-well plates for 5–6 days and then stimulated with 5 nM VIC, 2.5 μM TCZ, 5 μM TCZ, 5 nM VIC + 2.5 μM TCZ, 5 nM VIC + 5 μM TCZ, or 0.1% DMSO (control) for 5–6 days. Fresh medium containing the drug(s) was changed twice. Colony-forming assays were immediately performed using crystal violet staining after 10–12 days. Viable colonies were fixed with methanol, stained with 0.05% crystal violet for 20 min, washed with phosphate-buffered saline (PBS), and air-dried. Relative colonies were analyzed using an image analyzer. They were qualitatively observed and confirmed by performing at least two independent experiments.

### 4.5. Fluorescence-Activated Cell Sorting (FACS) Analysis

FACS analysis was performed as previously described [19,41,42]. Briefly, KBV20C cells were grown in 60-mm-diameter dishes and treated with 5 nM VIC, 2.5 μM TCZ, 5 μM TCZ, 5 nM VIC + 2.5 μM TCZ, 5 nM VIC + 5 μM TCZ, or 0.1% DMSO (control) for 1 day. Cells were then dislodged by trypsin and pelleted by centrifugation. Cell pellets were washed thoroughly with PBS, suspended in 75% ethanol for at least 8 h at 4 °C, washed with PBS, and resuspended in a cold propidium iodide (PI) staining solution (100 μg/mL RNase A and 50 μg/mL PI in PBS) for 30 min at 37 °C. These stained cells were analyzed in two independent experiments for relative DNA content using a Novocyte Flow cytometer (ACEA Biosciences, San Diego, CA, USA). They were qualitatively observed and confirmed by performing at least two independent experiments.

### 4.6. Annexin V Analysis

Annexin V analysis was conducted using an annexin V-fluorescein isothiocyanate (FITC) staining kit (BD Bioscience, Franklin, NJ, USA) as previously described [19,41,42]. Briefly, KBV20C cells were grown in 60-mm-diameter dishes and treated with 5 nM VIC, 60 nM eribulin, 0.5 μg/mL vinorelbine, 1 μg/mL docetaxel, 2.5 (or 5 μM) azole antifungal drugs, 2.5 (or 5 μM) rifabutin, alone or in combination with antimitotic drugs (VIC, eribulin, vinorelbine, or docetaxel), or 0.1% DMSO (control) for 24 h or 48 h. These cells were then dislodged using trypsin and pelleted by centrifugation. The cell pellet was washed with PBS. After adding 5 μL of annexin V-FITC and 5 μL of PI to cells in 100 μL of binding buffer, the mixture was then incubated at room temperature for 30 min. Stained cells were analyzed in two independent experiments using a Novocyte Flow cytometer (ACEA Bio-sciences, San Diego, CA, USA). They were qualitatively observed and confirmed by performing at least two independent experiments.

### 4.7. Rhodamine123 or Calcein-AM Uptake Tests

To assess the ability of a drug to inhibit P-gp, rhodamine 123 or calcein-AM uptake tests were performed as described previously [16,18,19,31]. Briefly, P-gp-overexpressing KBV20C or MCF-7/ADR cells were grown in 60-mm-diameter dishes and treated with 10 μM verapamil, 2.5 μM aripiprazole, 2.5 μM reserpine, or 5 μM azole antifungal drugs and then incubated at 37 °C for 1 h. After removing the medium, cells were washed with PBS. Cells stained with rhodamine 123 (0.5 μM) or calcein-AM (0.1 μg/mL) for 3 h were analyzed in two independent experiments using a Novocyte Flow cytometer (ACEA Biosciences, San Diego, CA, USA). They were qualitatively observed and confirmed by performing at least two independent experiments.

### 4.8. Western Blot Analysis

Total cellular proteins were extracted as described previously [18,19,31]. Briefly, cells grown in 60 mm dishes and treated with 5 nM VIC, 1 μM TCZ, 2.5 μM TCZ, 5 nM VIC + 1 μM TCZ, 5 nM VIC + 2.5 μM TCZ, or 0.1% DMSO (control) for 1 day. They were then washed twice with cold PBS and detached. For total protein isolation, cells were suspended in a PRO-PREP™ protein extract solution (iNtRON, Seongnam, Korea) and placed on ice for 30 min. The suspension was collected after centrifugation at 15,000× *g* for 15 min at 4 °C. Protein concentrations were measured using a protein assay kit (Bio-Rad, Hercules, CA, USA) according to the manufacturer’s instructions. Proteins were resolved by sodium dodecyl sulfate–polyacrylamide gel electrophoresis (SDS-PAGE) and subjected to Western blot analysis as previously described [28,29].

### 4.9. Statistical Analysis

All data are presented as mean ± S.D. from two independent experiments performed in triplicate. All statistical analyses were performed using one-way analysis of variance (ANOVA) followed by Bonferroni’s test. Analysis was performed using Graph Pad Prism Software Version 5.0 (GraphPad Software, San Diego, CA, USA). We indicated statistically significant sensitization effects following co-treatment based on a *p*-value of 0.01 (**), 0.001 (**), 0.001(***), or 0.0001 (****).

## Figures and Tables

**Figure 1 ijms-23-13809-f001:**
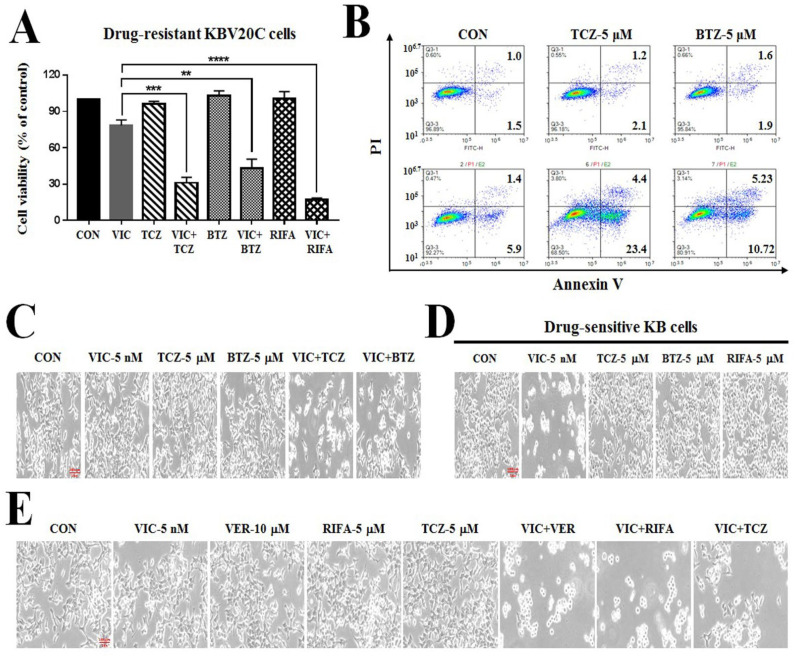
Both TCZ and BTZ increase VIC cytotoxicity in P-gp-overexpressing drug-resistant KBV20C cancer cells. (**A**) Drug-resistant KBV20C cells were plated on 96-well plates and grown to 30–40% confluence. The cells were then treated for 48 h with 5 nM VIC, 5 μM TCZ, 5 μM BTZ, 5 μM rifabutin (RIFA), 5 nM VIC with 5 μM TCZ (VIC + TCZ), 5 nM VIC with 5 μM BTZ (VIC + BTZ), 5 nM VIC with 5 μM rifabutin (VIC + RIFA), or 0.1% DMSO (CON). Cell viability assay was performed as described in “Materials and methods”. The data are presented as the mean ± S.D. of at least two experiments repeated in triplicate experiments. Data are presented as mean ± SD. ** *p* < 0.01, *** *p* < 0.001, or **** *p* < 0.0001 was considered be statistically significant. (**B**) KBV20C cells were treated with 5 nM VIC, the indicated μM concentrations of TCZ or BTZ alone, or in combination with 5 nM VIC, or 0.1% DMSO (CON). After 24 h, annexin V analyses were performed as described in Materials and Methods. (**C**) KBV20C were treated with 5 nM VIC, 5 μM TCZ, 5 μM BTZ, 5 nM VIC with 5 μM TCZ (VIC + TCZ), 5 nM VIC with 5 μM BTZ (VIC + BTZ), or 0.1% DMSO (CON). After one day, all cells were observed using an inverted microscope at ×100 magnification. (**D**) Drug-sensitive parent KB cells were treated with 5 nM VIC, 5 μM TCZ, 5 μM BTZ, 5 μM rifabutin (RIFA), or 0.1% DMSO (CON). After one day, all cells were observed using an inverted microscope at ×100 magnification. (**E**) KBV20C were treated with 5 nM VIC, 10 μM verapamil (VER), 5 μM TCZ, 5 μM BTZ, 5 nM VIC with 10 μM verapamil (VIC + VER), 5 nM VIC with 5 μM TCZ (VIC + TCZ), 5 nM VIC with 5 μM rifabutin (VIC + RIFA), or 0.1% DMSO (CON). After one day, all cells were observed using an inverted microscope at ×100 magnification.

**Figure 2 ijms-23-13809-f002:**
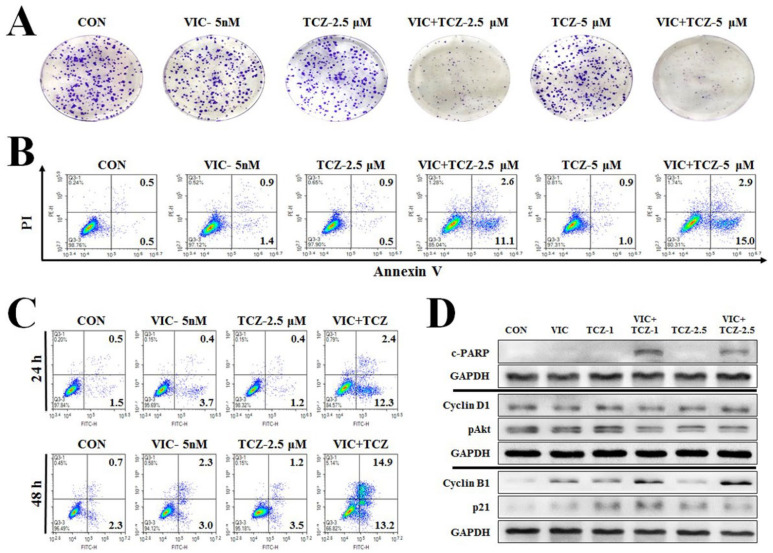
TCZ increases apoptosis in VIC-treated KBV20C cells. (**A**) KBV20C were treated for three days with 5 nM VIC, the indicated μM concentrations of TCZ alone, or in combination with 5 nM VIC, or 0.1% DMSO (CON). Colony-forming assays were immediately measured with crystal violet staining after total 10 days. (**B**) KBV20C cells were treated with 5 nM VIC, the indicated μM concentrations of TCZ alone, or in combination with 5 nM VIC, or 0.1% DMSO (CON). After 24 h, annexin V analyses were performed as described in Materials and Methods. (**C**) KBV20C cells were treated with 5 nM VIC, 2.5 μM TCZ alone, or in combination with 5 nM VIC, or 0.1% DMSO (CON). After 24 h or 48 h, annexin V analyses were performed as described in Materials and Methods. (**D**) KBV20C were treated with 5 nM VIC, 1 μM TCZ, 2.5 μM TCZ, 5 nM VIC with 1 μM TCZ (VIC + TCZ-1), 5 nM VIC with 2.5 μM TCZ (VIC + TCZ-2.5), or 0.1% DMSO (CON). After 24 h, western blot analysis was performed using antibodies against C-PARP, cyclin D1, pAkt, cyclin B1, p21, and GAPDH.

**Figure 3 ijms-23-13809-f003:**
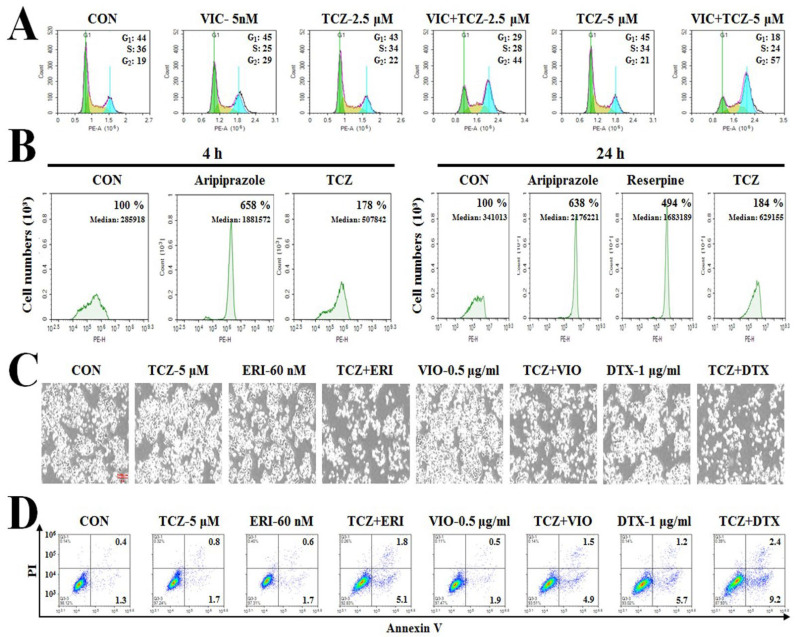
Co-treatment with TCZ increases cytotoxicity toward other antimitotic drug-treated KBV20C cells. (**A**) KBV20C were treated with 5 nM VIC, the indicated μM concentrations of TCZ alone, or in combination with 5 nM VIC, or 0.1% DMSO (CON). After 24 h, FACS analyses were performed as described in Materials and Methods. (**B**) KBV20C cells were treated with 2.5 μM aripiprazole, 2.5 μM reserpine, 5 μM TCZ, or 0.1% DMSO (CON). After 1 h (left panel) or 21 h (right panel), all cells were stained with rhodamine 123 for 3 h and examined by using FACS analysis as described in Materials and Methods. (**C**,**D**) KBV20C cells treated with 5 μM TCZ, 0.5 μg/mL vinorelbine (VIO), 60 nM eribulin (ERI), 1 μg/mL docetaxel (DTX) alone, or in combination with 5 μM TCZ, or 0.1% DMSO (CON). After 24 h, all cells were observed using an inverted microscope at ×40 magnification (**C**), or annexin V analyses (**D**) were performed.

**Figure 4 ijms-23-13809-f004:**
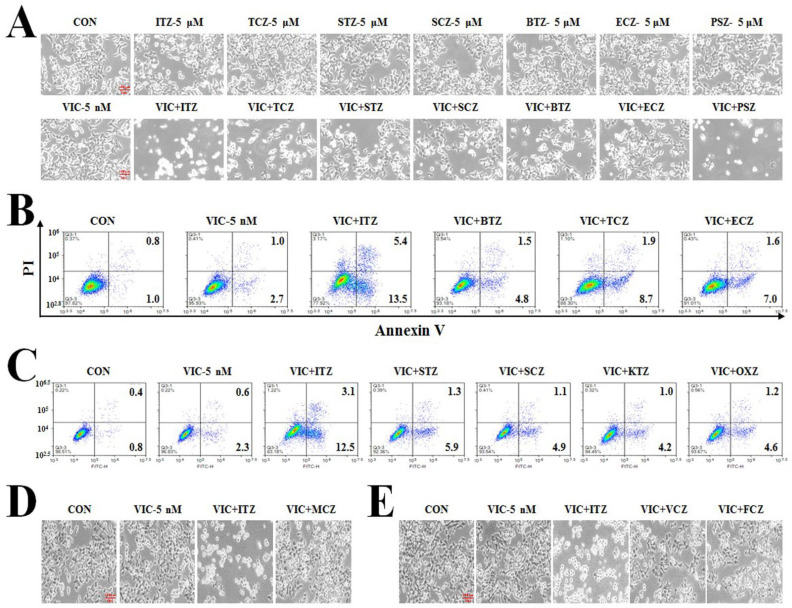
Low treatment with TCZ, itraconazole, and posaconazole exert higher cytotoxic effects in P-gp-overexpressing cancer cells than that of other azole antifungal drugs. (**A**) KBV20C were treated with 5 nM VIC, the indicated μM concentrations of itraconazole (ITZ), TCZ, sertaconazole (STZ), sulconazole (SCZ), BTZ, econazole (ECZ), or posaconazole (PSZ) alone, or in combination with 5 nM VIC, or 0.1% DMSO (CON). After one day, all cells were observed using an inverted microscope at ×100 magnification. (**B**,**C**) KBV20C were treated with 5 nM VIC, 5 nM VIC with 2.5 μM itraconazole (VIC + ITZ), 5 nM VIC with 2.5 μM BTZ (VIC + BTZ), 5 nM VIC with 2.5 μM econazole (VIC + ECZ), 5 nM VIC with 2.5 μM sertaconazole (VIC + STZ), 5 nM VIC with 2.5 μM sulconazole (VIC + SCZ), 5 nM VIC with 2.5 μM ketoconazole (VIC + KTZ), 5 nM VIC with 2.5 μM oxiconazole (VIC + OXZ), or 0.1% DMSO (CON). After 24 h, annexin V analyses were performed as described in Materials and Methods. (**D**,**E**) KBV20C were treated with 5 nM VIC, 5 nM VIC with 2.5 μM itraconazole (VIC + ITZ), 5 nM VIC with 2.5 μM miconazole (VIC + MCZ), 5 nM VIC with 2.5 μM voriconazole (VIC + VCZ), 5 nM VIC with 2.5 μM fluconazole (VIC + FCZ), or 0.1% DMSO (CON). After one day, all cells were observed using an inverted microscope at ×100 magnification.

**Figure 5 ijms-23-13809-f005:**
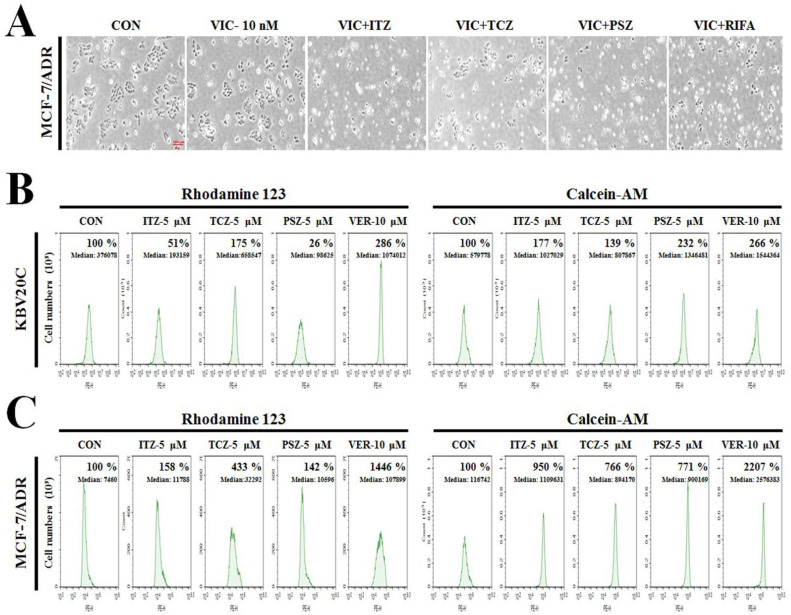
TCZ, itraconazole, and posaconazole exhibit high P-gp inhibitory activity in calcein-AM substrate efflux assays (**A**) P-gp-overexpressing drug-resistant MCF-7/ADR cells were treated with 10 nM VIC, 10 nM VIC with 5 μM itraconazole (VIC + ITZ), 10 nM VIC with 5 μM TCZ (VIC + TCZ), 10 nM VIC with 5 μM posaconazole (VIC + PSZ), 10 nM VIC with 5 μM rifabutin (VIC + RIFA), or 0.1% DMSO (CON). After two days, all cells were observed using an inverted microscope at ×100 magnification. (**B**) KBV20C cells were treated with 5 μM itraconazole (ITZ), 5 μM TCZ, 5 μM posaconazole (PSZ), 10 μM verapamil (VER), or 0.1% DMSO (CON). After 1 h, all cells were stained with rhodamine 123 (left panel) or calcein-AM (right panel) for 3 h and examined by using FACS analysis as described in Materials and Methods. (**C**) MCF-7/ADR cells were treated with 5 μM itraconazole (ITZ), 5 μM TCZ, 5 μM posaconazole (PSZ), 10 μM verapamil (VER), or 0.1% DMSO (CON). After 1 h, all cells were stained with rhodamine 123 (left panel) or calcein-AM (right panel) for 3 h and examined by using FACS analysis as described in Materials and Methods.

## Data Availability

Not applicable.

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
