# Peer review of "Terconazole, an Azole Antifungal Drug, Increases Cytotoxicity in Antimitotic Drug-Treated Resistant Cancer Cells with Substrate-Specific P-gp Inhibitory Activity"

_ijms, 2022, doi:10.3390/ijms232213809_

Round 1

Reviewer 1 Report

Ji Sun Li et al present the potential of Terconazole as an anticancer drug. This article would be of potential interest to the readership of IJMS as  Terconazole has recently been described as a potential anticancer approach in pancreatic cancer/ hemangioma/ prostate cancer or melanoma.  The authors propose a potential mechanism of action of the drug in cancer cell, as it would induce cell cycle arrest in P-glycoprotein overexpressing resistant cancer.

  While interesting, several points need to be improved:

- Fig 1: while looking for cell viability is legitimitate, this should be combined with a measure of cell death (such as propidium iodide)

- Fig 2a: while 2d colony formation assay is appropriate, this should be combined with a 3D culture model (ultralow attachment spheroids, or droplets) - that is a better model for drug testing

- Fig 2d:  cleaved PARP should be normalized to total PARP, phospho AKT to total AKT

- Fig 3- 5: please monitor cellular stress in the different conditions, for example by assessing oxidative stress/ or DNA damage.

Author Response

Ji Sun Li et al present the potential of Terconazole as an anticancer drug. This article would be of potential interest to the readership of IJMS as  Terconazole has recently been described as a potential anticancer approach in pancreatic cancer/ hemangioma/ prostate cancer or melanoma.  The authors propose a potential mechanism of action of the drug in cancer cell, as it would induce cell cycle arrest in P-glycoprotein overexpressing resistant cancer.

Response: We would like to thank the reviewer for these constructive comments. We have revised the manuscript based on the reviewers recommendation. If any inadequacies persist, please let us know.

  While interesting, several points need to be improved:

- Fig 1: while looking for cell viability is legitimitate, this should be combined with a measure of cell death (such as propidium iodide)

Response: We thank the reviewer for this critical comment. Given that we revealed cellular death using annexin V analysis in Figures 2B, 2C, and 4B, we omitted the PI results.

: In accordance with this suggestion, we added annexin V analysis for VIC+TCZ and VIC+BTZ co-treatments in Figure 1B, which indicate double staining with both PI and FITC-conjugated annexin V antibody.

: We also performed FACS analysis for VIC+TCZ and VIC+BTZ co-treatment. We have included the supplementary results below for your evaluation.

: Following the annexin V and FACS analysis, we conclude that low-dose TCZ exhibited greater cytotoxicity than BTZ (Fig. 1A, 1B, and below figure).

- Fig 2a: while 2d colony formation assay is appropriate, this should be combined with a 3D culture model (ultralow attachment spheroids, or droplets) - that is a better model for drug testing

Response: We appreciate the reviewer’s constructive comments. We agree that a 3D culture model (ultralow attachment spheroids or droplets) should be established in accordance with the reviewer’s suggestions. However, we would need to set up the methods in our laboratory, given that these techniques are yet to be established in our laboratory. In future studies, we will consider introducing these methods for investigating drugs that could sensitize P-gp overexpressing resistant cancer cells. In the revised manuscript, we have added the following sentence in the Discussion section: “Further assessment of sensitization using ultralow attachment spheroids or droplet assay will facilitate the rapid application of TCZ, especially in patients resistant to combination therapy with antimitotic drugs.”

- Fig 2d:  cleaved PARP should be normalized to total PARP, phospho AKT to total AKT

Response: We thank the reviewer for this comment. In our western blot analysis, we typically employed b-actin or GAPDH for normalization. We assumed that PARP and AKT protein levels are not active forms. We also assumed that the increase or decrease in respective protein levels did not indicate an increase or decrease in their active forms. For example, the cellular localization of PARP and AKT proteins is not proportionally representative of their active forms. Based on the suggestion, we will consider normalizing total protein levels of PARP and AKT to those of cleaved-PARP and phospho-AKT, respectively, in the future, on purchasing the relevant antibodies.  

: In the present manuscript, GAPDH was used to normalize WB results (Figures 2D) based on densitometry analysis of C-PARP and phospho-AKT levels. We have added the following statement in the revised manuscript: In the densitometric analysis, co-treatment resulted in 2-fold higher C-PARP levels than treatment with a single agent. In the densitometric analysis, p-AKT levels were similar between co-treatment and single agent treatments.

- Fig 3- 5: please monitor cellular stress in the different conditions, for example by assessing oxidative stress/ or DNA damage.

Response: We thank the reviewer for this important comment. We agree that the upregulation of DNA damage or autophagy should be determined as suggested by the reviewer. Using western blot analysis, we typically examine the upregulation or downregulation of key proteins (pH2AX or a-LC3B) involved in DNA damage or autophagy pathways. However, we failed to achieve consistent results with repeated analysis for relevant proteins. Therefore, we omitted the results and provided no discussion in the manuscript. In the revised manuscript, we have added the following statement in the Discussion section: “Future investigation should explore whether VIC+TCZ can increase autophagy or DNA damage in-depth mechanism studies.”

Reviewer 2 Report

In this original paper, the authors provide insight into developing new anti-cancer drugs. Besides, they focus on the advantages of novel compounds. This original work contains interesting aspects but needs some improvement for publication, especially in statistical analysis, results organization, and figure quality. My specific comments are:

In all figures where cells microscopy appears, needs to improve the image quality

Figure 1 Cell viability graphic does not indicate the bar comparison in the significance analysis to identify against what bar is compared. Please re-analyze the data and explain with a more extensive statistical analysis of the significance.

Cell microscopy lacks quality, does not point to any important image segment, or does not support any reported results.

Please support the results and conclusions with more extensive statistical analysis to establish the impact of the proposed new therapeutic drugs. 

Please number the lines in all manuscripts to optimize the review process 

Author Response

In this original paper, the authors provide insight into developing new anti-cancer drugs. Besides, they focus on the advantages of novel compounds. This original work contains interesting aspects but needs some improvement for publication, especially in statistical analysis, results organization, and figure quality. My specific comments are:

Response: We thank the reviewer for the constructive comments. We truly appreciate the detailed response and useful suggestions that improve our study.

: We have made necessary revisions to the manuscript based on these recommendations. If any further changes are required, please do let us know. Thank you for your consideration, and we look forward to your response.

In all figures where cells microscopy appears, needs to improve the image quality

Response: We thank the reviewer for this comment. We realized that the images appear dark and altered the intensity of black and white of all photomicrographs. In addition, we altered colony-forming assay results considering the image intensity.

Figure 1 Cell viability graphic does not indicate the bar comparison in the significance analysis to identify against what bar is compared. Please re-analyze the data and explain with a more extensive statistical analysis of the significance.

Response: We thank the reviewer for this comment. In accordance with your suggestion, we re-analyzed the data shown in Figure 1A using ANOVA and t-tests. We also changed the Figure graph with a bar comparison. In addition, we indicated statistically significant sensitization effects following co-treatment, based on a p-value of 0.01 (**), 0.001 (**), 0.001(***), or 0.0001 (****).

Cell microscopy lacks quality, does not point to any important image segment, or does not support any reported results.

Response: We thank the reviewer for these critical comments. In the revised manuscript, we altered photomicrographs to exhibit better intensity by improving the brightness.

: As indirectly stated, microscopic observations are not quantitative assays. In our microscopic analysis, we confirmed the results of at least two independent tests. Typically, we present microscopic findings whenever cell numbers significantly differ between control and drug-treated groups. Considering the significantly different cell densities between control and drug-treated groups, the representative figures are unbiased captured results. We added the following statement in the Materials and Methods section: They were qualitatively observed and confirmed by performing at least two independent experiments.

: To compare control and drug-treated cells, we occasionally count cell numbers in a square area when establishing differences using only microscopic observations appears challenging. In all Figures and repeated microscopic results in this manuscript, we confirmed the differences using microscopic observations, as the differences between control and drug treatment were visually apparent. We will consider the reviewers comment in future analyses for microscopic observations

: We also emphasized that microscopic results were confirmed using tetrazolium-based assay (MTS), FACS, and annexin V analysis.

: The viability assay (tetrazolium-based assay; MTS) was not an effective technique for short-term observation (24 h), whereas microscopic observations were markedly effective in observing drug-induced effects over 24 h. Therefore, we compared both techniques (viability assay and microscopic observations) while presenting the results. After comparing the viability assay and microscopic observations, we determined the effectiveness of drugs in the first screening process and subsequently examined the underlying sensitization-related molecular mechanisms.

Please support the results and conclusions with more extensive statistical analysis to establish the impact of the proposed new therapeutic drugs. 

Response: We thank the reviewer for these critical comments.

: The results for annexin V, FACS, and rhodamine (calcein-AM) assays were values from representative results of two independent experiments. Therefore, statistical evaluation could not be undertaken for these results. We confirmed similar trends between two independent experiments. We added the following statement in the Materials and Methods section: They were qualitatively observed and confirmed by performing at least two independent experiments.

: Moreover, we did not consider statistical analysis for these results, as differences between single agent (or control) and co-treatment were qualitatively significant. In the revised manuscript and study, instead of comparing values for the above-listed assays, we attempt to determine G2 arrest, apoptosis, or P-gp inhibition through qualitative analysis.

: In accordance with your suggestion, we realized that we should have performed three independent experiments to determine statistical significance. This will be considered in our further studies. In the present manuscript, we would like to conclude that VIC+TCZ co-treatment increased P-gp inhibition, G2 arrest, and apoptosis, as determined using qualitative analysis

Please number the lines in all manuscripts to optimize the review process 

Response: Thank you for noting this omission. We used the template file for the initial submission and failed to realize the missing line numbers. The necessary changes have been made in the revised manuscript.

Round 2

Reviewer 1 Report

The authors corrected most points of concern. 

Reviewer 2 Report

Please check the spelling in a few segments of the manuscript.